# Exploring the Multifunctional Roles of Odontoglossum Ringspot Virus P126 in Facilitating Cymbidium Mosaic Virus Cell-to-Cell Movement during Mixed Infection

**DOI:** 10.3390/v13081552

**Published:** 2021-08-06

**Authors:** Shu-Chuan Lee, Hsuan Pai, Ying-Wen Huang, Meng-Hsun He, Yun-Lin Song, Song-Yi Kuo, Wen-Chi Chang, Yau-Heiu Hsu, Na-Sheng Lin

**Affiliations:** 1Institute of Plant and Microbial Biology, Academia Sinica, Taipei 11529, Taiwan; shuchuanlee@gate.sinica.edu.tw (S.-C.L.); chuppadeju904@hotmail.com (H.P.); mhhe@gate.sinica.edu.tw (M.-H.H.); as0200594@gate.sinica.edu.tw (Y.-L.S.); 2Graduate Institute of Biotechnology, National Chung Hsing University, Taichung 40227, Taiwan; ywhuang0101@gmail.com (Y.-W.H.); unchatbleu124@gmail.com (S.-Y.K.); yhhsu@nchu.edu.tw (Y.-H.H.); 3College of Biosciences and Biotechnology, Institute of Tropical Plant Sciences, National Cheng Kung University, Tainan 70101, Taiwan; sarah321@mail.ncku.edu.tw

**Keywords:** odontoglossum ringspot virus (ORSV), cymbidium mosaic virus (CymMV), viral synergism, RNA silencing, viral suppressor of RNA silencing (VSR), *Phalaenopsis*

## Abstract

Synergistic interactions among viruses, hosts and/or transmission vectors during mixed infection can alter viral titers, symptom severity or host range. Viral suppressors of RNA silencing (VSRs) are considered one of such factors contributing to synergistic responses. Odontoglossum ringspot virus (ORSV) and cymbidium mosaic virus (CymMV), which are two of the most significant orchid viruses, exhibit synergistic symptom intensification in *Phalaenopsis* orchids with unilaterally enhanced CymMV movement by ORSV. In order to reveal the underlying mechanisms, we generated infectious cDNA clones of ORSV and CymMV isolated from *Phalaenopsis* that exerted similar unilateral synergism in both *Phalaenopsis* orchid and *Nicotiana benthamiana*. Moreover, we show that the ORSV replicase P126 is a VSR. Mutagenesis analysis revealed that mutation of the methionine in the carboxyl terminus of ORSV P126 abolished ORSV replication even though some P126 mutants preserved VSR activity, indicating that the VSR function of P126 alone is not sufficient for viral replication. Thus, P126 functions in both ORSV replication and as a VSR. Furthermore, P126 expression enhanced cell-to-cell movement and viral titers of CymMV in infected *Phalaenopsis* flowers and *N*. *benthamiana* leaves. Taking together, both the VSR and protein function of P126 might be prerequisites for unilaterally enhancing CymMV cell-to-cell movement by ORSV.

## 1. Introduction

Mixed infections by plant viruses are commonly found in nature and cause many important viral diseases [1]. The virus–virus and viruses–host interactions in co-infection scenarios may be antagonistic or synergistic depending on the combinations of viral strains, hosts, infection time-points and, in some cases, insect vectors [1,2]. The synergistic interactions arising from mixed infection may result in beneficial effects for one or all viral partners, potentially increasing viral titers and enhancing viral movement and/or symptoms in the host plants [3,4,5,6,7,8]. The pathological consequences of mixed viral infections are rather unpredictable, but may involve infection time-lapses beyond a threshold level [2]. The molecular mechanisms underlying viral synergism remain largely unclear. However, virus-encoded suppressors that counteract the plant RNA silencing surveillance system not only regulate gene expression for proper development in eukaryotes [9] but also serve in a major strategic role against virus infection [10,11]. Indeed, a few studies have shown that viral suppressors of RNA silencing (VSRs) are determinants of viral synergism. For instance, in overexpressing the VSRs of potyviral helper component-proteinase (HC-Pro) [8,12], sweet potato chlorotic stunt virus RNase III [13], cucumber mosaic virus 2b [14] and tobacco mosaic virus (TMV) P126 replicase [15] increased host susceptibility and the accumulation of unrelated viruses.

Orchids are the largest Monocotyledon family, with more than 900 genera and 20,000 to 35,000 species [16]. More than 50 viruses have been reported to infect orchids [17]. Among them, the odontoglossum ringspot virus (ORSV) and cymbidium mosaic virus (CymMV) are two of the most prevalent orchid-infecting viruses worldwide [18,19]. Single infection with ORSV or CymMV causes orchids to display growth defects and diminished flower quality, whereas co-infection results in synergistic effects that exacerbate disease severity, which can erode the orchid’s economic value [18,19,20,21,22] and even cause plant death [18,23,24].

ORSV and CymMV are taxonomically distinct viruses exhibiting different genome organizations and morphologies. ORSV is a *Tobamovirus*, with a genome size of ~6.6 kb and a tRNA-like structure at its 3′-terminus [25]. This single-stranded, positive-sense RNA genome has four open reading frames (ORFs) that encode P126 replicase and its readthrough product P183, movement protein (MP) and capsid protein (CP). The mutation of Phe^50^ to Ser^50^ in P126/183 of an ORSV infectious cDNA clone abolished ORSV replication [26]. Moreover, the ORSV replicase is responsible for host range restriction in *Nicotiana sylvestris* [27]. For other tobamoviruses, P126 was shown to act in a non-membrane-bound form as an VSR [28,29], but it may form a membrane-bound heterodimer with P183 for viral replication. P126 has methyltransferase (MET), non-conserved region (NON) and helicase (HEL) domains. The MET, HEL and a portion of the NON-domains of TMV P126 individually possess VSR activity and, notably, the VSR activities of the MET and HEL domains can operate independently from their enzymatic activities [30]. The P126 protein interferes with the RNA silencing pathway by binding to duplexes of small interfering RNAs (siRNAs), which may inhibit HEN1-mediated siRNA methylation and potentially sequesters them to prevent the RNA-induced silencing complex (RISC) formation [31,32]. By contrast, CymMV is a *Potexvirus*, with a poly(A)-tailed, single-stranded and positive-sense RNA genome of ~6.2 kb [33]. The CymMV genome comprises five ORFs that encode RNA-dependent-RNA polymerase (RdRp) for genome replication, three overlapping triple-gene-block proteins (TGBps) for cell-to-cell and long-distance movement and CP for virus encapsidation. The CymMV CP also contributes to viral movement within different hosts [34].

Synergistic ORSV and CymMV isolated from *Cattleya* exhibited enhanced replication upon co-infection of *Dendrobium* petal protoplasts relative to the single infection, displaying earlier and higher viral RNA titers [3]. Interestingly, transgenic plants expressing CymMV TGBps supported the cell-to-cell movement of a movement-deficient ORSV mutant, and the cell-to-cell movement-deficient CymMV was rescued in ORSV MP transgenic plants [6]. However, long-distance movement of a CymMV CP-deficient mutant could be rescued in ORSV CP transgenic plants, but CymMV CP transgenic plants were unable to restore systemic movement of an ORSV CP-deficient mutant [6]. Although *Phalaenopsis* orchids co-infected with ORSV and CymMV exhibited significant symptom intensification, unlike the viral synergism found in *Cattleya*, CymMV movement was unilaterally boosted by ORSV in co-inoculated *Phalaenopsis* [35]. Virus titers of CymMV and ORSV were similar upon single or co-infection of protoplasts from *Nicotiana benthamiana* leaves and *Phalaenopsis amabilis* flowers [35,36].

Here, we show that ORSV P126 is a VSR and our mutation analysis demonstrates that its VSR activity can operate independently of its virus replication activity. Moreover, the VSR and protein functions of ORSV P126 are required in order to facilitate CymMV cell-to-cell movement, enabling the unilateral synergism that enhances CymMV infection. Our findings provide insights into the mechanism underlying asymmetric synergism during CymMV and ORSV co-infection in *Phalaenopsis*.

## 2. Materials and Methods

### 2.1. Plants and Growth Conditions

*N. benthamiana* and *C**henopodium quinoa* plants were grown at 28 °C with a 16 h/8 h light/dark photoperiod in a walk-in growth chamber. *Phalaenopsis* plants were purchased from Clone International Biotech Co. Ltd. (Pingtung, Taiwan). *Phalaenopsis* plants were first validated to be CymMV-free and ORSV-free by reverse transcription-polymerase chain reaction (RT-PCR) [37] before being subjected to virus inoculation.

### 2.2. RT-PCR Detection of CymMV and ORSV

CymMV-free and ORSV-free *Phalaenopsis* plants were screened by multiplex RT-PCR by using primers for the amplification of CymMV, ORSV and an internal *nad5* control (Appendix A) [37]. Briefly, the RT-PCR first-strand cDNA reaction was performed by using SuperScript™ III Reverse Transcriptase (Invitrogen, Carlsbad, CA, USA) and via simultaneous addition of three reverse primers: CymMV CP-R1, ORSV CP-R1 and mt-R1. PCR was then performed on cDNAs by using a multiplex primer set (including 2.5 μM mt-F2/mt-R1, 1.25 μM CymMV CP-F1/CymMV CP-R1 and 1.25 μM ORSV CPF1/ORSV CP-R1) and 2× SuperRed PCR Master Mix (Biotools Co. Ltd., New Taipei City, Taiwan) by following the manufacturer’s instructions. The RT-PCR products were then analyzed by agarose gel electrophoresis.

### 2.3. Virus Isolation, Purification, Inoculation and RNA Extraction

CymMV and ORSV isolated from diseased *Phalaenopsis* plants were maintained in *N. benthamiana* plants [35,36], and the virions were purified as described previously [38]. Purified virions were adjusted to 2 mg/mL and stored at −80 °C.

For single virus inoculation, plants were inoculated with 0.5 μg/leaf of CymMV or ORSV virions or with 1 μg of viral transcripts. For co-infection scenarios, a mixture of 0.5 μg/leaf CymMV and 0.5 μg ORSV virions was used to inoculate leaves of 3-week-old *N. benthamiana*, 5-week-old *C. quinoa* or 6-month-old acclimatized *Phalaenopsis*. The replication of the ORSV infectious clone OS4 and its P126 mutants was verified by inoculation of viral transcripts to *C. quinoa* or agro-infiltration of *Agrobacterium* carrying pkn, pkOS4, pkOS4-P126R, pkOS4-P126S or pkOS4-P126R to *N. benthamiana*.

Plant leaf total RNAs were extracted by TRIzol^®^ Reagent following the manufacturer’s instructions (Invitrogen, Carlsbad, CA, USA). RNA quality and quantity were assessed using a NanoDrop 1000 Spectrophotometer (ThermoFisher Scientific Inc., Wilmington, DE, USA) and samples were stored at −80 °C.

### 2.4. Plasmid Construction

The full-length CymMV and ORSV genomes were directly amplified by RT-PCR from the total RNAs of orchids inoculated with wild isolate (wt) virions of CymMV and ORSV. Briefly, one microgram of total RNA was used for first-strand cDNA synthesis by using the SuperScript III reverse transcriptase (Invitrogen, Carlsbad, CA, USA) and primer ORSV-RP1 or CymMV-RP1, respectively. The first strand cDNAs were then used as templates for primer pairs—ORSV-FP1 and ORSV-RP1 for ORSV; CymMV-FP1 and CymMV-RP1 for CymMV—to synthesize the full-length genomes by PCR using Phusion Flash High-Fidelity PCR Master Mix (ThermoFisher Scientific Inc.). The amplified fragments were purified by using a QIAEX II Gel Extraction Kit (QIAGEN, Hilden, Germany), digested with *Sac*I and *Sma*I (for CymMV) or *Bam*HI and *Spe*I (for ORSV) and then cloned into pU119 vector at the corresponding sites to generate pUCy1 and pUOS4 constructs, respectively.

The full-length ORSV cDNA clones harboring the P126 M1104R/S/A mutation—pUOS4-P126R, pUOS4-P126S and pUOS4-P126A—were generated by using a QuikChange Site-Directed Mutagenesis Kit (Agilent Technologies, Santa Clara, CA, USA) and the primer pairs that are presented in Appendix A. The full-length Cy1 fragment of pUCy1 was further PCR-amplified and subcloned into pkn binary vector [39] to generate pkCy1 plasmid. The same procedure was used to generate pkOS4, pkOS4-P126R, pkOS4-P126S and pkOS4-P126A from pUOS4, pUOS4-P126R, pUOS4-P126S and pUOS4-P126A, respectively. Clones were confirmed by sequencing. The CymMV construct with a green fluorescent protein (GFP) reporter, pkCy1GFP, was generated by inserting the GFP gene downstream of the duplicated CymMV CP subgenomic promoter (−100 to +22 [40]). A 35S promoter-driven mCherry expression cassette was further amplified by PCR and inserted into the *Bgl*II site of pkn vector to obtain pkn::mCherry. The pkCy1GFP::mCherry construct was generated by subcloning the 2× 35S promoter-Cy1GFP expression cassette from pkCy1GFP into pkn::mCherry.

For transient expression assays, viral ORFs or mutants were PCR-amplified from pUCy1, pUOS4 or pUOS4 mutants and cloned between two *Sma*I sites or between the *Sma*I and *Bam*HI sites of the pBIN61 binary vector [41] with a HA sequence at their C-terminus. The ORSV P126 ORF mutants were generated by using a QuikChange Site-Directed Mutagenesis Kit (Agilent Technologies), with pBIN61-ORSV P126 and pBIN61-ORSV P126-R as templates. All primers used for cloning and detection are listed in Appendix A. All clones described below were verified by restriction enzyme digestion and DNA sequencing. The VSR indicator, pBIN61-GFP (GFP) and the positive controls pBIN19-P19 and pBIN61-P25, which encode tomato bushy stunt virus (TBSV) P19 and potato virus X (PVX) P25 proteins, were gifts from Dr. David Baulcombe, University of Cambridge, UK.

### 2.5. In Vitro Transcription

The plasmids of pUCy1 or pUOS4 were linearized by *Sma*I or *Spe*I digestion, respectively. Capped transcripts corresponding to the wt virus were synthesized through in vitro transcription by using T7 RNA polymerase (Promega, Madison, WI, USA) in the presence of the cap analogue m7G(5′)ppp(5′)G (New England Biolabs, Inc., Ipswich, MA, USA) under the reaction conditions recommended by the manufacturer.

### 2.6. Tissue Blotting and RNA Blot Analysis

Tissue blotting to reveal virus infection and distribution in inoculated *Phalaenopsis* leaves was performed as described previously [35,42]. In brief, leaf slices from inoculated and adjacent non-inoculated tissues were printed onto Hybond™-N^+^ membranes (GE Healthcare Life Sciences, Buckinghamshire, UK) and hybridized with riboprobes specific to CymMV/ORSV full-length CP genes by using the DIG Nucleic Acid Detection Kit (Sigma-Aldrich, St. Louis, MO, USA). Northern blot analysis was performed, as described previously [43,44], by using DIG-labeled probes targeted to the CP and 3′ UTR sequence of the pUCy1 and pUOS4 genomes.

### 2.7. Agrobacterium Infiltration and GFP Imaging

The VSR activity assay for individual viral ORFs was performed as described previously [45] with some modifications. *Agrobacterium tumefaciens* harboring corresponding plasmids was grown overnight at 28 °C. The bacteria were then pelleted down and suspended in 0.1 volume of induction solution (10 mM MgCl_2_ and 100 μM acetosyringone in sterile water) and incubated in the dark at room temperature for 4 hours. Equal volumes of *A. tumefaciens* cultures (OD_600_ = 1) expressing positive sense-GFP, pBIN61-eGFP and *A. tumefaciens* cultures (OD_600_ = 1) harboring pBIN61-viral ORF-HA expression vectors were mixed and co-infiltrated into the leaves of 3-week-old to 4-week-old *N. benthamiana*. Expression plasmids for the VSRs P19 encoded by TBSV (pBIN19-P19) and P25 encoded by PVX (pBIN61-p25) were used as positive controls, whereas empty vector (pBIN61) served as a negative control. The agroinfiltrated leaves were illuminated under a long-wavelength UV lamp (Black Ray model B 100 AP) and photographed at 4 days post-agroinfiltration (DPA). The GFP signal intensity was detected and measured by using an IVIS Lumina III LT in vivo Imaging System (XENOGEN Co., Alameda, CA, USA) [46]. All experiments were repeated three times. The images were processed electronically by using Adobe Photoshop CS5.

In order to measure CymMV cell-to-cell movement, cultures of *A. tumefaciens* carrying pBIN61 (vector), pBIN61-P126 or P126-derived mutants were induced and adjusted to OD_600_ = 0.5 for infiltration. Two hours after infiltration, the bacterial ooze of *A. tumefaciens* carrying pkCy1GFP::mCherry was pinpricked into infiltrated leaves using a toothpick for infection. CymMV cell-to-cell movement was observed by using an IVIS Lumina III LT in vivo Imaging System (XENOGEN Co., Alameda, CA, USA) at 6 DPA and then quantified by using its Live Imaging software. Images were processed in ImageJ (NIH).

### 2.8. Western Blot Analysis

The leaf tissue from infiltration zones was ground in liquid nitrogen and resuspended (20 (*v/w*)) in 2× MURB buffer (100 mM sodium phosphate, pH 7.0, 50 mM MES, 2% (*w/v*) sodium dodecyl sulfate, 6 M Urea; and 1 mM sodium azide with freshly added 10% (*v/v*) β-mercaptoethanol). The protein samples were incubated at 55 °C for 15 min and then subjected to SDS-PAGE and immunoblot analysis. Anti-CymMV CP serum was used to detect CymMV, as described previously [17]. The detection of ORSV was performed by using anti-ORSV CP serum generated by immunizing rabbits with purified ORSV virions. The overexpression of ORSV P126 and derived mutant proteins were detected via their C-terminus HA tags by using the monoclonal anti-HA antibody produced in mice (Sigma-Aldrich, St. Louis, MO, USA). Plant actin was used as a protein loading control and was detected by using monoclonal anti-Actin (clone 10-B3, Sigma, St. Louis, MO, USA) and horseradish peroxidase (HRP)-linked secondary antibodies. GFP was detected using HRP-conjugated anti-GFP antibody (Abking Biotechnologies Inc., Taipei, Taiwan). Immunoreactivity was detected by using the Clarity™ Western ECL Substrate (Bio-Rad Laboratories, Inc., Hercules, CA, USA) as per the manufacturer’s instructions. Blots were exposed to X-ray film for various time-periods. All experiments were repeated three times. The signal intensity was quantified in ImageJ.

### 2.9. Statistical Analysis

Statistical analysis was performed by one-way ANOVA and followed by a post-hoc test. The significant differences are marked as follows: *: *p* < 0.05; **: *p* < 0.01; ***: *p* <0.001; ****: *p* < 0.0001; and ns: no significant difference.

## 3. Results

### 3.1. Construction of ORSV and CymMV Infectious cDNA Clones

The viral synergism caused by ORSV and CymMV isolated from *Phalaenopsis* [35,36] was confirmed by back-inoculation of purified virions to *Phalaenopsis* plants alone or mixed (Appendix A). In order to characterize the biological properties of *Phalaenopsis* ORSV and CymMV isolates, we generated their infectious cDNA clones. By aligning all eight full-length ORSV genome sequences currently available from the NCBI database, we used the conserved 5′ and 3′ sequences to design the primers ORSV-FP1 and ORSV-RP1 (Appendix A) for the direct amplification of the complete ORSV genome from total RNA of infected *Phalaenopsis* leaves by RT-PCR. The amplified ORSV cDNA genome was linked to an upstream T7 promoter and a downstream restriction enzyme site (*Spe*I) (Appendix A and Appendix A). *Spe*I cleavage resulted in the correct 3′ terminus sequence of ORSV, thereby preventing the incorporation of any non-viral sequences that might adversely affect ORSV clone infectivity [47,48]. Infectivity of ORSV cDNA clones was verified by inoculating the in vitro synthesized ORSV RNA transcripts into *C. quinoa*. Inoculated *C. quinoa* leaves presented localized chlorotic lesions in which ORSV accumulation was detected by Western blot using anti-ORSV CP serum after 4 days post-inoculation (DPI) (Appendix A). The infectious transcripts derived from ORSV cDNA clone, denoted OS4 hereafter, were inoculated into *C. quinoa* for virion purification. The CPs purified from OS4 or ORSV wild isolate (OR-wt) virions are ~17 kDa and are indistinguishable from one another (Appendix A). The OS4 genome sequence comprises 6611 nucleotides in length and demonstrates 97–99% identity relative to the ORSV genomes published in the NCBI database, and they display similar genome organizations (Appendix A).

In order to generate CymMV infectious cDNA clones, we used the same strategy as for ORSV. The full-length CymMV genome was amplified using specific primers CymMV-FP1 and CymMV-RP1 and linked to an upstream T7 promoter and a downstream poly(A) tail; we designated this infectious cDNA clone as pUCy1 (Appendix A and Appendix A). Inoculation of in vitro synthesized pUCy1 RNA transcripts, denoted Cy1 hereafter, into *C. quinoa* leaves resulted in tiny lesions, which displayed detectable CymMV CP accumulation based on Western blot using anti-CymMV CP serum after 8 DPI (Appendix A). The size of the CP protein from Cy1 virions purified from inoculated *C. quinoa* was ~25 kDa, i.e., indistinguishable from that of wild CymMV isolate (Cy-wt) (Appendix A). Sequence analysis of pUCy1 revealed it was 6225 nucleotides long, excluding the poly(A) tail, and it exhibits 96–97% identity to most CymMV genomes published in the NCBI database and presents similar genome organization (Appendix A).

### 3.2. Asymmetric Synergism of CymMV and ORSV Infection in N. benthamiana and Phalaenopsis Orchid

Since OS4 and Cy1 derived from infected *Phalaenopsis* were shown to be infectious (Appendix A), we assayed synergistic interactions between OS4 and Cy1 in an experimental host, *N. benthamiana*. The infection of Cy1 alone resulted in tiny localized chlorotic/white lesions in the inoculated leaves (IL) of *N. benthamiana* at 14 DPI, but few lesions or no symptoms on systemic leaves (SL) were observed (Figure 1A). The growth of Cy1-inoculated *N. benthamiana* was comparable to mock-inoculated plants (Figure 1A). Accumulation of Cy1 CP could be detected in IL at 14 DPI and slightly in SL via Western blot using an antibody against CymMV CP (Figure 1B). RNA blotting confirmed Cy1 RNA accumulation in inoculated plants (Figure 1C). By contrast, OS4 did not cause visible symptoms on IL, but plants showed yellowing and distortion of SL and whole-plant growth defects at 14 DPI (Figure 1A). As for Cy1, there was a substantial accumulation of OS4 CP and RNA in both IL and SL based on Western blot and RNA blot analyses, respectively (Figure 1B,C). However, *N. benthamiana* co-infected with Cy1 and OS4 exhibited more severe symptoms in IL, SL, and whole-plant growth defects than presented upon single infection (Figure 1A). In co-infected leaves, Cy1 CP and Cy1 RNA were hyper-accumulated in IL relative to levels observed for single infection, indicating that Cy1 levels were enhanced by OS4 (Figure 1B,C). In SL, dramatic increases in Cy1 CP and Cy1 RNA levels were noted upon co-infection (Figure 1B,C). However, the levels of OS4 CP and RNA were similar regardless of single or co-infection (Figure 1B,C), which is consistent with our previous reports [35,36]. Thus, our infectious clones Cy1 and OS4 faithfully recapitulated the asymmetric synergistic effects of virion wild isolates from *Phalaenopsis* orchid, i.e., unilaterally enhanced CymMV accumulation and systemic movement due to ORSV, which result in enhanced disease symptoms during co-infection.

In order to further examine the viral synergism of Cy1 and OS4 in a natural host, we inoculated Cy1 and/or OS4 virions onto half a leaf-tip of *Phalaenopsis* leaves. After 10 DPI, no symptoms were observed for leaves inoculated solely with Cy1 or OS4 (Appendix A), although tissue blots revealed accumulations of Cy1 and OS4 RNAs, respectively (Appendix A). However, in contrast to symptomless single infection, co-infected orchid leaves presented necrotic, ring-like lesions (Appendix A) and enhanced Cy1 movement relative to non-inoculated neighboring tissues at 10 DPI (Appendix A), which is consistent with our findings from *N. benthamiana* (Figure 1). Thus, the Cy1 and OS4 faithfully represent the biological characteristics of Cy-wt and OR-wt from the *Phalaenopsis* orchid (Appendix A).

### 3.3. Identification of ORSV-Encoded P126 as a VSR

Several studies have shown that expression of VSRs can increase host susceptibility and virus titers of unrelated viruses [2,12,13,14,15]. Accordingly, we wondered if ORSV also encodes a VSR that could contribute to CymMV and ORSV synergism. We identified candidate ORSV VSRs by co-expressing GFP together with individual OS4-encoded proteins in an *Agrobacterium*-mediated expression system [45]. Two well-characterized VSRs, TBSV P19 and PVX P25 (P19 and P25 hereafter) were used as strong or moderate VSR controls, respectively, and an empty vector acted as a negative control (Figure 2A). The OS4 ORFs or protein domains, including the MET, NON, HEL and polymerase domain p54, were individually cloned into binary vector pBIN61 to generate transient-expressing protein constructs (Figure 2B). After the co-expression of GFP and individual HA-tagged viral proteins, the relative GFP signal intensity was measured at 4 DPA. As anticipated, the GFP signal was enhanced strongly by P19 and moderately by P25 (Figure 2C,D). Among the ORSV constructs we tested, only P126 showed statistically significant VSR activity, which was comparable to P25 (Figure 2C,D). Moreover, P126 expression also increased the GFP protein accumulation as P19 and P25 did (Figure 2E). These results indicate that the ORSV P126 replicase is a VSR. When the viral proteins were co-expressed, all OS4 viral proteins, including P126 (Figure 2E), were detectable by Western blots, with the exception of the HEL and p54 proteins (Appendix A). Thus, their VSR activities need to be examined further. In addition, we also assayed the potential VSR candidates among Cy1 ORFs. Although Cy1 RdRp, TGBp1, TGBp3 and CP were indeed expressed (Appendix A), none of them exhibited significant VSR activity (Appendix A). As Cy1 TGBp2 protein was not detected, its VSR activity could not be concluded.

### 3.4. Single Amino Acid Substitution Abolishes ORSV P126 VSR Activity

The VSR activity of several *Tobamovirus* P126 mutants has previously been reported to be diminished or abolished [28,29,30]. We substituted several amino acids to alanine (A)—including Gly (G)^325^, Leu (L)^348^, Lys (K)^367^, Glu (E)^601^, Glu (E)^663^ and Val (V)^742^—which corresponded to tobamovirus replicase mutants exhibiting defective VSR activity (Appendix A). We also mutated Met (M)^1104^ to Arg (R), since a screening of ORSV cDNA clones had revealed this mutation eliminated ORSV replication. Several single (e.g., E^601^A) and double (e.g., E^601^A and M^1104^R) mutants were generated (Appendix A). Most of the ORSV P126 single mutants exhibited a similar level of VSR activity to parental P126 protein in terms of enhancing GFP accumulation, except for P126-R^1104^ (P126R) (Figure 3 and Appendix A). A further two single mutants, P126S and P126A representing the mutation of M^1104^ to Ser or Ala, respectively, exhibited similar VSR activity (Figure 3A,C) to P126, despite P126A or P126S mutant protein levels being reduced relative to P126 (Figure 3B). Notably, the P126R protein level was barely detectable and that could be due to the protein instability (Figure 3B and Appendix A). As expected, none of the P126R double mutants presented any VSR activity (Appendix A).

### 3.5. ORSV P126 VSR Activity Uncouples ORSV Replication

Since the full-length ORSV cDNA clone harboring the P126R mutation was unable to replicate, we wondered if the ORSV clones with the P126S and P126A mutations, which retained VSR activity, remained infectious. We generated the corresponding cDNA clones OS4-P126S and OS4-P126A and inoculated their transcripts or OS4 control onto *C. quinoa* leaves. We observed localized lesions at 5 DPI upon OS4 control inoculation but not for the P126R, P126S or P126A mutants (Figure 4A). Moreover, only OS4 control-inoculated tissues exhibited substantial CP accumulation, whereas none of leaves inoculated with the mutants did (Figure 4B). The replication of OS4 and OS4-P126 mutants were further assayed on *N. benthamiana* by agro-infiltration. The accumulations of CP and viral RNAs were detected from OS4 but not from OS4-P126 mutants by Western and RNA blots at 2 and 8 DPA, respectively (Figure 4C,D), indicating that the amino acid substitution in P126 M^1104^ abolished ORSV replication. Thus, P126 has two distinct roles in virus replication and RNAi suppression.

### 3.6. ORSV P126 Is the Synergistic Determinant That Enhances CymMV Cell-to-Cell Movement

Our previous studies demonstrated that the co-infection of *Phalaenopsis*-derived CymMV and ORSV isolates did not enhance viral accumulation at the single cell level [35,36]. However, we observed greatly enhanced accumulation of Cy1 in *N. benthamiana* IL and SL upon co-infection with OS4 (Figure 1B,C). In order to test if ORSV P126 synergistically enhances CymMV accumulation by increasing CymMV cell-to-cell movement, we generated a dual expression cassette, pkCy1GFPF::mCherry, that is driven by the 35S promoter for *Agrobacterium*-mediated transient expression (Figure 5A). The mCherry and GFP signals, representing the infection foci of *Agrobacterium* and Cy1GFP, respectively, could be observed simultaneously after agro-infiltration of *N. benthamiana* leaves with *Agrobacterium* carrying pkCy1GFP::mCherry (Appendix A). At 4 DPA, we observed that the mCherry signal was restricted to the infection site, whereas the expanded GFP signal represented cell-to-cell movement of Cy1GFP (Appendix A).

In order to evaluate the function of P126 protein in CymMV cell-to-cell movement, we overexpressed P126 protein or vector control in *N. benthamiana* leaves, followed by agro-infection of Cy1GFP::mCherry by using a toothpick (Figure 5B). Expanded Cy1GFP signal could be observed at 6 DPA (Figure 5C) and the virus spreading was evaluated by measuring the relative signal areas of GFP/mCherry. By comparing the spread of Cy1GFP signal under P126 expression relative to vector control, we found that Cy1GFP cell-to-cell spread was enhanced by the expression of P126 to 1.41-fold (Figure 5D). Furthermore, the total detected GFP signals were used to evaluate the Cy1GFP titer of the infection foci. The results showed that the expression of P126 increased Cy1GFP titer to 2.18-fold compared with Vec (Figure 5E), indicating that P126 overexpression stimulates both cell-to-cell movement and viral titers. Moreover, similar results were also observed while assayed in *Phalaenopsis* flowers (Appendix A).

In order to further assay if any other VSRs have such enhancement for CymMV movement, TBSV P19 and PVX P25 were included in this assay. Despite that P19 and P25 have strong and moderate VSR activities, respectively, only TBSV P19 stimulated CymMV cell-to-cell movement (1.28-fold) and accumulation (2.63-fold) as P126 did, but P25 could not (Figure 5D,E). Interestingly, although P19 showed significantly stronger VSR activity than P126 (Figure 2C), its enhancement of CymMV cell-to-cell movement was inferior to P126 (Figure 5D).

Taken together, our findings demonstrate that the ORSV P126 expression alone was sufficient to increase CymMV accumulation by enhancing CymMV cell-to-cell movement. The P126S and P126A mutants retain comparable VSR activity relative to parental protein but lacked replicase activity. Thus, we conclude that ORSV P126 has multiple roles in virus infection, including viral replication, VSR and promoting CymMV movement during co-infection scenarios.

## 4. Discussion

Synergistic interactions among two or more viruses have complex consequences that can vary depending on the host, viral strains and infection timing among other factors [1,2]. ORSV and CymMV isolates from *Cattleya* orchids mutually enhanced viral RNA replication in orchid protoplasts upon mixed infection [3]. However, this synergistic interaction is restricted in *Phalaenopsis* orchids to a unilateral enhancement of CymMV movement rather than mutually bolstered replication [35,36]. These studies imply that divergences in synergistic interactions between ORSV and CymMV could be attributable to different virus isolates and orchid species. Here, the infectious ORSV and CymMV cDNA clones we generated from *Phalaenopsis*-derived isolates (termed OS4 and Cy1, respectively) not only displayed similar biological activities to wild-isolated viruses, but also recapitulated the synergism-induced phenotypes manifested upon co-infection, including symptom enhancement and increased CymMV titers in both natural (*Phalaenopsis*) and experimental (*N. benthamiana*) hosts. Moreover, titers and movement of Cy1 were enhanced by OS4 but not vice versa (Figure 1 and Appendix A). These outcomes indicate that our Cy1 and OS4 clones preserved all the functional features of viral single-infection and co-infection and that the unilateral benefits relative to Cy1 accumulation might be driven by factors and/or effects arising from OS4 co-infection.

VSRs not only facilitate viral infection by counteracting the RNA silencing element of plant immunity [10,49,50] but also contribute to synergism during mixed infections. Herein, we have identified the ORSV P126 replicase as a VSR (Figure 2), which is consistent with the replicases of other tobamoviruses [28,29]. Notably, despite potexvirus TGBp1 (P25) being a well-known VSR [28,29,51], none of the CymMV viral proteins we tested exhibited any significant VSR activity, not even CymMV TGBp1 (Appendix A).

In order to test if the VSR activity of P126 is required for ORSV replication, we generated several P126 mutants (Figure 4). One such mutant, P126R, lacked VSR activity, most likely due to impaired protein stability given its low protein levels (Figure 3B and Appendix A). However, P126 mutants lacked virus replication ability in *N. benthamiana*, whether they exhibited VSR activity (Figure 3 and Figure 4). Thus, the VSR activity of the ORSV P126 replicase is not sufficient to support the successful viral infection in plants. Similar results have been reported for mutations that disrupted the helicase domain motifs of TMV 126/183 kDa proteins in that they inhibited virus replication but did not affect RNAi suppression [52]. Importantly, most host factors identified to date as interacting with tobamovirus P126 protein target the HEL domain in order to facilitate tobamovirus multiplication and pathogenicity [53]. Whether amino acid change at residue M^1104^ alters interactions with host factors required for replication warrants further investigation. Since ORSV P183, which is the readthrough protein of P126, is consequently affected by P126 mutation, a role for it in replication deficiency cannot be excluded.

We have shown that the expression of OS4 P126 alone effectively promoted cell-to-cell movement and accumulation of Cy1 in inoculated leaves (Figure 5), indicating that one of the key underlying mechanisms of CymMV and ORSV synergism is dependent on promoting movement and accumulation of CymMV by ORSV P126. The P25, with comparable VSR activity as P126 (Figure 2C), however, did not enhance CymMV movement at all (Figure 5D). Interestingly, P19 displayed stronger VSR activity than P126 but showed relatively less efficiency than P126 to stimulate CymMV cell-to-cell movement (Figure 2C and Figure 5D). This may be attributed to the function of P19 in increasing host susceptibility and resulting in increased accumulation of unrelated viruses [54]. Thus, the efficient promotion for viral synergism might be related to the compatible or specific interaction between virus and VSR protein rather than just VSR activity alone. Other viral VSRs, such as potyvirus P1/HC-Pro, crinivirus RNaseIII and cucumovirus 2b, have been characterized as mediating synergistic effects by increasing virus titers, long-distance movement and/or pathogenicity in addition to suppressing host RNA silencing responses [2,8,12,13]. Moreover, VSRs from a wide spectrum of viruses, including animal viruses, were demonstrated to complement cell-to-cell movement of a VSR-depleted turnip crinkle virus [55], but a role for TMV P126 in facilitating cell-to-cell movement of other viruses has yet to be revealed. Nevertheless, it has been shown that TMV P126 and/or P183 proteins are involved in the cell-to-cell movement of TMV [56,57]. Furthermore, intracellular trafficking of P126 alone or TMV viral replication complexes harboring P126 required the association of TMV P126 to microfilaments [58,59], suggesting that ORSV-promoted CymMV movement might also be modulated directly or indirectly via P126-microfilament interactions. The CymMV and ORSV viruses isolated from *Cattleya* orchids exhibited mutual replication enhancement [3], but *Phalaenopsis*-derived isolates only unilaterally enhanced CymMV movement [35,36]; this highlights the complexities and different layers of regulation between virus–virus and virus–host interactions.

## Figures and Tables

**Figure 1 viruses-13-01552-f001:**
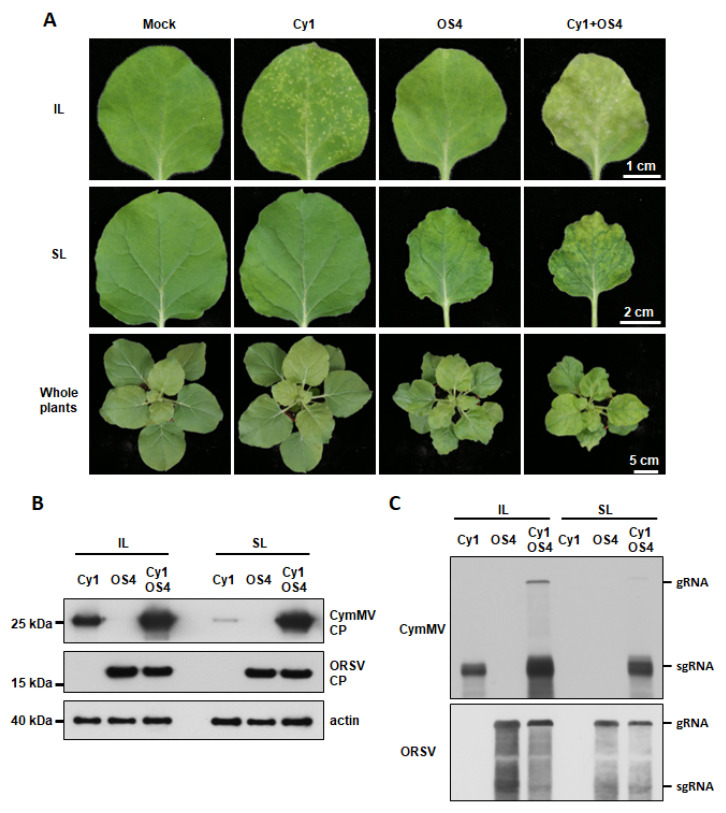
Cy1 and OS4 infectivity and synergism in inoculated *N. benthamiana* plants. (**A**) Symptoms and synergistic effects upon co-infection with Cy1 and OS4 virions. Eighteen-day-old *N. benthamiana* plants were individually inoculated with 0.5 μg/leaf of Cy1 or OS4 virions or co-inoculated with 1 μg/leaf of Cy1 and OS4 (0.5 μg/leaf each). Symptoms on inoculated leaves (IL), systemic leaves (SL) and whole plants were photographed at 14 DPI. (**B**) Immuno-detection of Cy1 and OS4 CPs from leaf extracts of inoculated plants at 14 DPI using antibodies against the CP of CymMV or ORSV. Actin was detected by using anti-Actin monoclonal antibody as a protein loading control. (**C**) RNA blots for assessing Cy1 and OS4 viral RNA accumulation. Total RNAs were extracted from leaves as in (**B**). RNA was detected by using DIG-labeled probes against the CP and 3′ UTRs of Cy1 or OS4, respectively.

**Figure 2 viruses-13-01552-f002:**
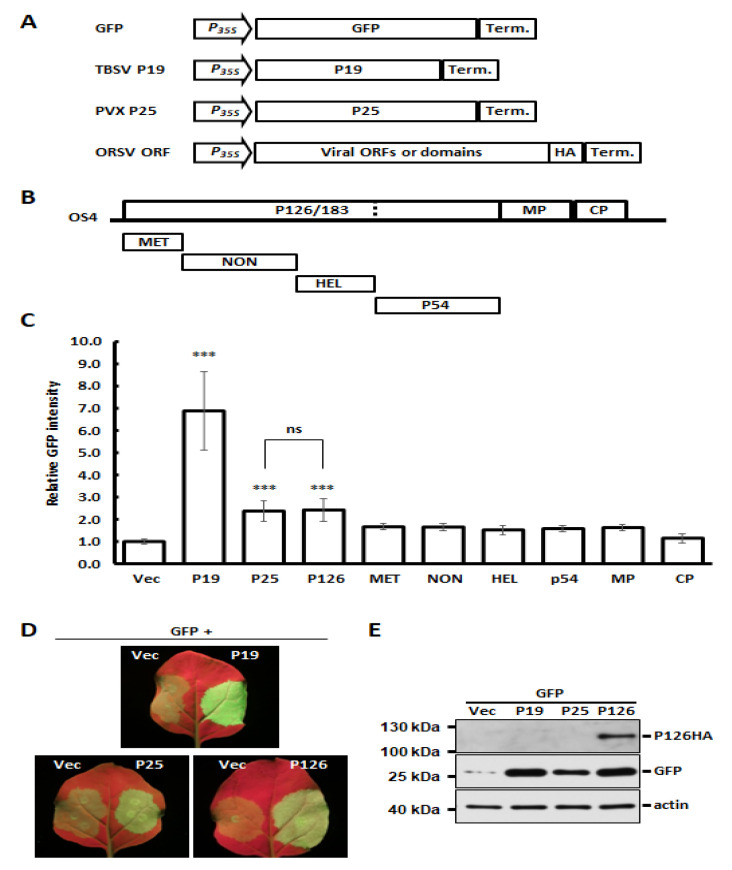
VSR assays of ORSV-encoded proteins and protein domains. (**A**) Schematics of expression cassettes for GFP (reporter), TBSV P19 and PVX P25 (VSR positive controls), as well as tested ORSV ORFs. (**B**) Individual ORFs and domains used for VSR assay. (**C**) Quantitative analysis of VSR activities. Overnight cultures of *Agrobacterium* carrying a corresponding plasmid were adjusted to final OD = 1. The co-infiltration of *N. benthamiana* leaves was performed by mixing *Agrobacterium* carrying pBIN61-GFP and *Agrobacterium* carrying plasmid positive for VSR or tested viral ORFs in a ratio of 1:1. GFP signals were detected and quantified using the IVIS Lumina III LT in vivo Imaging System at 4 DPA. Relative GFP intensity represents the mean of three independent experiments and was normalized to vector control. The significance was calculated by one-way ANOVA followed by post-hoc test and marked when *p* < 0.001 (***); ns: no significant difference. (**D**) Representative images of GFP signals as observed under hand-held UV lamp. (**E**) Accumulation of HA-tagged P126, GFP and actin in (**D**) was assessed by Western blotting by using antibodies against HA, GFP and actin, respectively.

**Figure 3 viruses-13-01552-f003:**
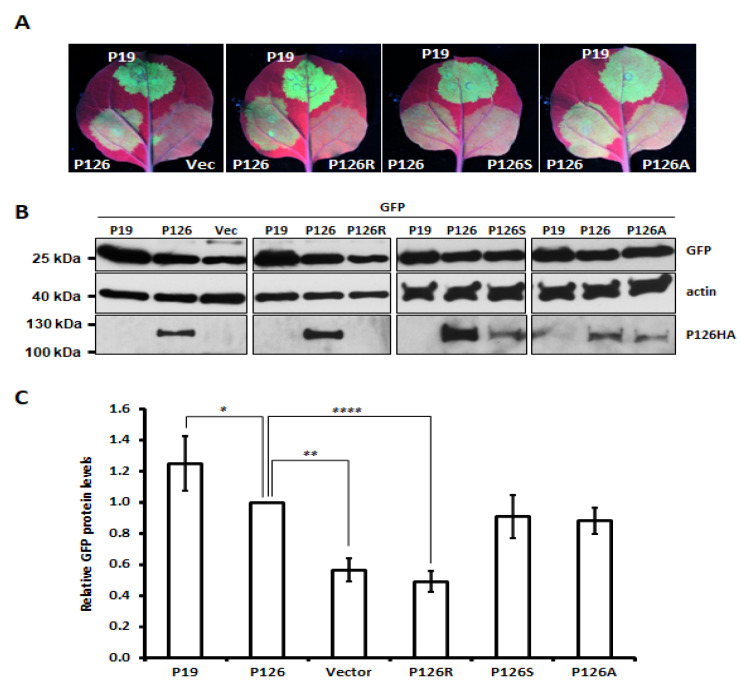
VSR activity of single amino acid substitution mutants of ORSV P126. (**A**) GFP signals upon co-expression with vector control (Vec), P19, P126 and derived P126 mutants. (**B**) Total protein was extracted from the infiltrated leaf discs and then subjected to Western blot in order to detect GFP, actin, P126 and mutant variants. (**C**) Relative GFP protein level was normalized with actin. The GFP level upon co-expression with P126 was set as 1. Vec: vector control. Data represent results from three different experiments. Significant differences are marked as follows: *: *p* < 0.05; **: *p* < 0.01; **** *p* < 0.0001 (one-way ANOVA followed by a post-hoc test).

**Figure 4 viruses-13-01552-f004:**
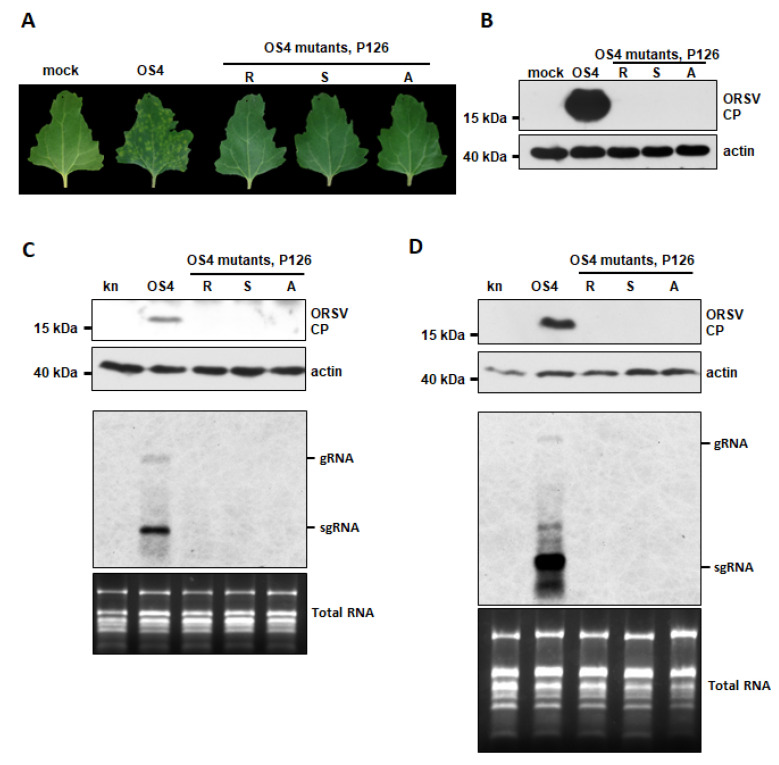
Replication of OS4 and its P126 mutants. (**A**) Symptoms of *C. quinoa* leaves inoculated with transcripts of OS4 and P126 mutants at 5 DPI. Mock represents leaves inoculated with phosphate buffer, which were used as a control. (**B**) Accumulation of ORSV CP in (**A**) as assessed by Western blot. Actin acted as a loading control. (**C**,**D**) Accumulations ORSV CP and viral RNAs in *N. benthamiana* at 2 DPA (**C**) and 8 DPA (**D**), respectively. *Agrobacterium* carrying pkn, pkOS4, pkOS4-P126-R, P126-S and P126-A mutants were adjusted to OD_600_ = 1 and infiltrated into 17-day-old *N. benthamiana* leaves. Western blot detection of ORSV CP (upper two panels) and RNA blot for ORSV RNAs (lower two panels) were performed as described in Figure 1. Total RNA was used as RNA loading control.

**Figure 5 viruses-13-01552-f005:**
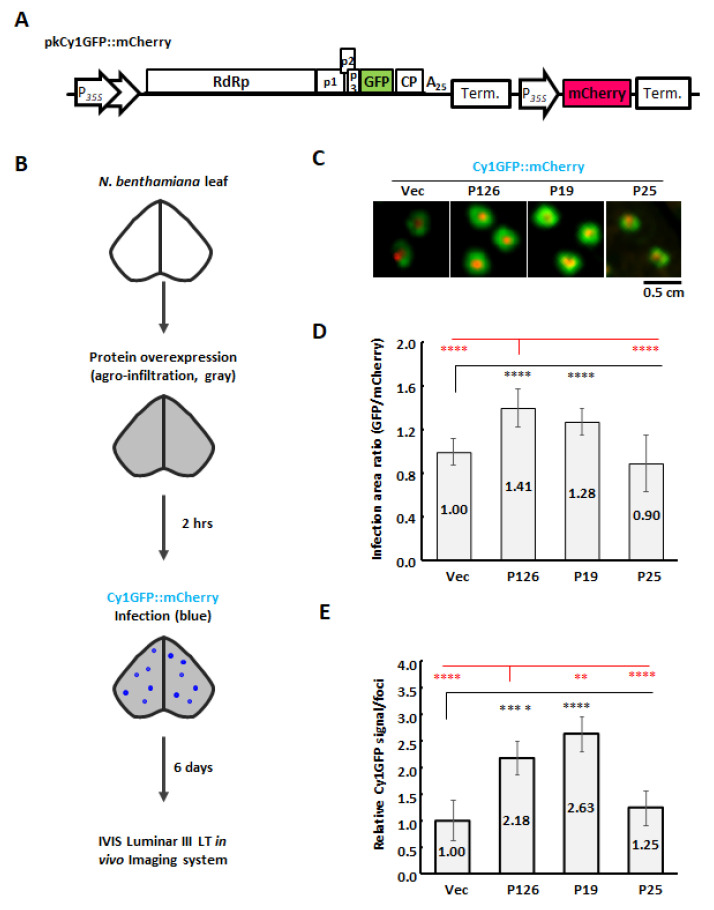
CymMV cell-to-cell movement and accumulation upon overexpressing VSRs in *N. benthamiana* leaves. (**A**) Schematic construct of pkCy1GFP::mCherry. (**B**) Flowchart for protein expression and CymMV inoculation. *Agrobacterium* carrying pBIN61 (Vec), pBIN61-P126 (P126), pBIN19-P19 (P19) or pBIN61-P25 (P25) were adjusted to OD_600_ = 0.5 and then infiltrated into 17-day-old *N. benthamiana* leaves. After 2 hours post-agroinfiltration, the *Agrobacterium* ooze carrying pkCy1GFP::mCherry was collected on a toothpick and pinpricked onto the infiltrated leaves. (**C**) Representative images of *Agrobacterium* primary infection sites (mCherry in red) and Cy1GFP spreading infected cells (GFP in green) using the IVIS Lumina III LT in vivo Imaging System at 6 DPA. Images were processed in ImageJ. (**D**) Quantification of CymMV movement. According to (**C**), the ratio of coverage areas was measured by the expanded Cy1GFP signals (GFP) against primary infection site (mCherry). (**E**) Effects of VSRs on accumulation of Cy1GFP. The relative Cy1GFP signals were measured by the total GFP signals per infection foci and quantification was normalized to vector control (black star) or P126 (red star). Data represent three experimental replicates, each of which encompassed analysis of six infection foci. Significant differences are marked as follows: **: *p* < 0.01 and ****: *p* < 0.0001 (one-way ANOVA followed by post-hoc test).

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
