# Peer review of "Exploring the Multifunctional Roles of Odontoglossum Ringspot Virus P126 in Facilitating Cymbidium Mosaic Virus Cell-to-Cell Movement during Mixed Infection"

_viruses, 2021, doi:10.3390/v13081552_

Round 1
Reviewer 1 Report
The authors of this manuscript (Viruses-1296607-v1) have added new data and have made the modifications required of its previous version to support their conclusions. They have also replied to each comment made of the previous version to the satisfaction of this reviewer. Thus, I am satisfied with the changes made.
There are a few word changes necessary, largely for better expression of the points being made, as follows:
- ln 120. Change “verified” to “verified”.
- ln 181. Define “DPA” here (as “days post-agroinfiltration”) at first use, rather than on ln 288-289.
- In Figs. S4A and S4C, change “nts” to “nt” – unit abbreviations are always given in the singular rather than the plural.
- ln 400. Delete either “that” or “but”. They are not compatible together as written here.
- ln 401. Rephrase to “…ORSV P126 has multiple roles in virus infection, including viral replication, VSR and promoting CymMV movement…”.
- ln 423. Delete the inserted “except TGBp2”, since it did not accumulate and therefore could not display activity. Also, as inserted, it implies that TGBp2 did show VSR activity. [It might be that TGBp2 requires TGBp1 or TGBp3 for stability, but that is not the issue here.]
Reviewer 2 Report
Comments Lee et al.,
In the manuscript presented by Lee et al., the authors present data supporting a multifunctional character of ORSV P126 exemplified by its role as VSR and by promoting the cell-to-cell movement of CymMV. The authors rely on the use of both host and experimental plant species to carry out the experiments, the results obtained are clear and conclusive, and they show evidences of the degree of complexity of the virus-virus and virus-plant interactions. This work may be of interest to the scientific community and especially for management programs against viral infections in species of commercial interest such as orchids. However, this reviewer has some comments to be considered by the authors, as indicated below.
Major comments:
- Section 3.4. Single amino acid substitution abolishes ORSV P126 VSR activity (and Figure 3). Correct me if I am wrong, because maybe I missed something or did not understood well, but as I understand without an accumulation of P126R the authors cannot conclude that the loss of the VSR activity of P126 is due to the P126R mutation. It could be the case that there is no VSR activity simply because there is no protein rather than the effect of the point mutation. Therefore, this section of the results should be reconsidered and rewritten, including the title of the epigraph so as not to fall into such an overstatement. Same as in line 426-427. Can the authors show western blots for the other replicates in order to show what is described in e.g. line 311 as "barely detectable protein". Otherwise, it is hard for me to believe that the protein was even detected at least once.
- For a comparison of more than two group means the one-way analysis of variance (ANOVA) is the appropriate method instead of the t test followed by a post-hoc test. Therefore, the authors have to redo the statistical analyzes. Same applies for Figure 3C, Figure 5D-E, and Figure S6C (and the corresponding Material and Methods section).
- The cited references are a bit outdated. There are more recent publications, especially those regarding VSRs, that deserve to be considered.
Minor comments:
- Line 119: OS4 stands for…?
- Line 120: maybe should be verified or validated instead of verified
- Line 187: can the authors describe how the bacterial ooze is obtained?
- Line 296: “Although,”, remove the comma.
- Line 293-298: Could the authors suggest possible causes why certain viral proteins have not accumulated enough? Have the authors checked if the transcripts were being accumulated properly? If so, please include this information.
Author Response
Please see the attachment.

This manuscript is a resubmission of an earlier submission. The following is a list of the peer review reports and author responses from that submission.
Round 1
Reviewer 1 Report
In this submission, the authors first described the construction of biologically active cDNA clones of ORSV (an orchid-infecting tobamovirus) and CymMV (an orchid-infecting potexvirus), the RNA transcripts of which could infect Chenopodium quinoa (Fig. 1), Nicotiana benthamiana (Fig. 2) and Phalaenopsis orchids (Fig. S4). Previously, this group and others have shown that that these two viruses can replicate in N. benthamiana and can interact synergistically in this dicot-host species, as well as in their natural monocot hosts, orchids. Here, they used their infection viral RNA transcripts to confirm the synergistic interaction in N. benthamiana (Fig. 2) and Palaenopsis orchids (Fig. S4). The authors then went on to verify that the P126 replication-associated protein of ORSV had viral RNA-silencing suppression (VSR) activity, as established for other tobamoviruses (Fig. 3). They also tested whether other ORSV proteins (P54 replicase-associated protein; MP; or CP), or domains of the P126 protein (‘MET’, ‘NON’ and ‘HEL’) had VSR activity, and found that none of the other ORSV or P126 protein domains had such activity (Fig. 3); the latter was in contrast to observations made by others with some of the P126 protein domains of TMV. They tested various mutants of P126, largely in the NON domain, but also near the C-terminus of the P126 – amino acid 1104, with a change of M to R, S, or A. While the M1104R mutant lost its VSR activity (Fig. 4), none of the other mutants lost their VSR activity; i.e., neither the other two 1104 mutants (M->A or M->S) (Fig. 4; Fig. S5), nor the three tested NON mutants (Fig. S5). However, none of the three P126 1104 mutant viruses could replicate in C. quinoa, based on accumulation of the ORSV CP (Fig. 5). Agroinfiltration and expression of P126 or the three P126 1104 mutants, followed by pin-pricking the agroinfiltrated zones with Agrobacterium in an ‘ooze’ (presumably from another plant previously agroinfilrated with Agrobacterium) containing a GFP-expressing CymMV cDNA clone in the Agrobacterium vector, along with a separate gene on the same plasmid expressing mCherry, were done to determine whether the P126 and its replication-deficient mutants (some of which still retained VSR activity) could stimulate the cell-to-cell movement of CymMV (Fig. 6). The authors found that the P126 mutant protein did not accumulate to detectable levels (by western blotting) in this assay, but that the WT P127 protein did so, and concluded that the latter could enhance the cell-to-cell movement of CymMV. They also concluded that both VSR and replication functions of ORSV P126 are necessary for the synergism functions.
There are a number of problems with this study. The major issues are given first, while other issues are given later.
- ln 304 and Fig. 2C. In contrast to previous publications examining either the VSR activities of various proteins or protein domains, here, the expression of only WT P126 was verified, with none of the P126 domains MET, NON, HEL, the other ORSV proteins (P54, MO or CP), or the various CymMV proteins assessed for VSR activity, was verified by western blot, to confirm expression and stable accumulation of these proteins. Thus, the authors cannot state categorically that these domains did not have VSR activities. Moreover, in Fig. 4B and Fig. S6, P126 mutant P126R showed much lower levels of accumulation than WT p126, while mutant P126S also showed some reduction in accumulation but no change in VSR activity, which the authors acknowledge on ln 330-336, and also concluded that the loss of VSR activity for mutant P126R was due to protein instability. The same may be the case for the P126 domains, the P54 domain in Fig. 2C, as well as the various CymMV protein assays for VSR activity, not shown. The results for P126 mutant stability also vary in different assays, with greater differences seen between WT P126 and one or all mutants in Figs. 4B, 6E and S6.
- If the P126 mutants are unstable and do not accumulate, they may still show some transitory VSR activity, even if virus accumulation does not occur. This is shown in Fig. S5B (where the data are not shown quantified), as well as in Fig 4C, where the M1104A and M1104S mutants of P126 show VSR effects that are not considered significantly different from WT P126. By contrast, the movement assay shows no accumulation of P126 mutant proteins (Fig. 6E), but still shows differences in the mean of the infection area or GFP accumulation (Figs. 6D and F) for mutants M1104A and M1104S vs. M1104R. These are considered not significantly different from the vector control, but in the case of the M1104A mutant, the data are also not significantly different from the WT P126 either! In Fig. 6, there is considerable standard error especially for the M124A mutant and to a lesser extent for the M1124S mutant, and even for the WT P126 samples in Fig. 6F. These variations all make it very difficult to draw any conclusions from these assays.
- The assay for replication of the P126 mutants (Fig. 5B) is flawed. If the mutants could still replicate the genomic RNAs of the virus, but could not make subgenomic RNAs to allow MP and CP synthesis, then they would still retain some replication function, but would not facilitate a progressive infection. Thus, detection of the CP is not sufficient to rule out a replication protein role or other P126 role in synergism for this protein.
- On ln 21-22, 81-83, 89, 207-209, 411-418, and 462-465, the authors are fixated on an issue not addressed in this work; i.e., whether or not infection of either Dendrobium protoplasts are Cattleya plants result in a synergism in which both viruses increase in concentration and not just the potexvirus, as seen in N. benthamiana, Phalaenopsis and C. quinoa. Given that this fixation is present in the Abstract, which should only refer to work being examined and work non-germane to the subject of the study, the reader would expect the authors to examine this issue in the study. That it is mentioned in the Introduction and further is raised again in the Results, Discussion and concluding remarks is even odder. These should be removed from this manuscript and saved for a future one.
Based on these data, I do not feel that the authors have made the case for their conclusions. More data are needed.
Other comments:
- The first two figures are validating the system developed, and confirming previous work. As such, these should be supplemental figures.
- Table 1, a long list of oligonucleotides used for cloning and mutagenesis, occupies 1.5 pages and is not necessary to understand the text. As such, it should be supplemental information.
- ln 2-3, 19-20, 35, 51-53, 56, 62, 70-71, 76, 297, 323 and 457. Taxonomy issues. The authors are not using the correct orthography of virus names. Only the taxonomic (conceptual) names of viruses are written in italics with a capital first letter for the first word of the name. The names of actual viruses causing disease or having other physical properties such as infection, disassembly, translation, replication, encapsidation, movement, transmission, purification, or (TEM) visualization, are written in Roman type in all lower case letters, unless part of a proper name, which is usually limited to animal viruses names after places. Also, since 2014, the ICTV has stated that taxonomic names of viruses cannot be abbreviated; the standard abbreviations only apply to the names of real viruses. In addition, the use of the family name as part of the species name (ln 51-53), has never been accepted by the ICTV. Moreover, abbreviations for any term, including virus names, can only be used if the term is used three times in the text, which is not the case here for ‘CMV’ and ‘SPCSV’. Finally, TMV was defined and abbreviated twice (ln 52-53 and ln 70-71).
- ln 62 and 76. The unit is ‘kb’, not ‘Kb’.
- ln 77 vs. Inconsistent abbreviation. On ln 77, RdRP was used, while in Table 1 (numerous entries, left and right columns) and in Fig. 1, RdRp was used. The latter is the preferred biochemical abbreviation.
- ln 201, 220, 232, 257, 261, 319, 340 and 357. The ‘western’, in ‘western blot’, should not have a capital ‘W’. Only ‘Southern’ (in Southern blots) has a capital first letter, since this was named after the person who developed it.
- ln 222. Redundancy. Delete ‘proteins’; after ‘CP’ and add an ‘s’ to ‘CP’ (‘CPs’).
- Fig. 2C requires the labelling of the bands observed, based on size markers.
- ln 297. The abbreviation PVX and TBSV should be defined at first use (ln 154).
- Fig. 3C. The ‘ns’ (‘no significance’) line is in the wrong place, being above the level of P25 and equal to the level of P126.
- ln 328 and Fig. 5. Figures have to be called out (first cited) in numerical order. Here, Fig. 5 was cited before Fig. 4 (ln 330). This problem is also the case for other supplemental figures: calling out Fig. S3 panels (ln 226, 230, 237) occurred before Fig. S2 was called out (ln 423).
- ln 417. The words ‘Cattleya orchids by Wong’s group’ was typed in a different font.
- ln 430-435. Again, this could be due stability differences which were not examined here.
Reviewer 2 Report
The manuscript by Lee et al examines the synergism between odontoglossum ringspot virus and cymbidium mosaic virus. In the title and throughout the manuscript, virus names should not be capitalized and italicized unless the authors are referring to the virus species. Virus names should not be italicized when referring to a physical virus.
In Figure 3, the entire leaf should be shown for all three VSR assays, not just for P19. P19 should and does show very strong VSR activity. It is difficult to judge the strengths of the other VSRs without the negative controls. This is evident in Figure 4 where P19 shows much stronger VSR activity than P126.
The authors constructed and assayed three single-amino-acid substitution mutants for VSR activity of ORSV P126. When the mutant proteins accumulated at much lower levels than the wild-type protein the authors concluded that the lack of VSR activity of the mutants could result from protein instability, which made it impossible to assess the roles of the amino acid positions in silencing suppression. Because of the apparent instability of the proteins and the lack of replication of viruses containing the mutations, it is difficult to see how the authors concluded that “VSR activity of the ORSV P126 replicase is essential but not sufficient to support successful viral infection in plants”.
As the authors described, previous studies showed that ORSV enhanced the cell-to-cell movement of CymMV. Figure 6 presents experiments to evaluate the possible role of ORSV P126 in enhancing CymMV movement. Because the P126 mutants accumulate very low levels of protein compared to wild-type P126 (see Figure 6 E), they are not informative on the question of the function of P126 and could be removed from the figure. That leaves a comparison of infection area between the vector control and wild-type P126 (relative accumulation of Cy1-y1-GFP is not a measure of virus movement). The GPF signals from treatments with P126 appear much brighter than those treated with the vector alone, and much brighter than the results presented in Figure 4. It seems possible that the apparent larger area could be just a brighter GFP signal. To control for that possibility, the authors should have included a VSR that would not be expected to enhance CymMV movement, for example either the P19 or P25 plasmids used in Figure 3. In their present form, the results are not strong enough to support the conclusion that ORSV P126 enhances CymMV cell-to-cell movement.